# Time-Dependent Changes in Hepatic Sphingolipid Accumulation and PI3K/Akt/mTOR Signaling Pathway in a Rat Model of NAFLD

**DOI:** 10.3390/ijms222212478

**Published:** 2021-11-19

**Authors:** Klaudia Sztolsztener, Karolina Konstantynowicz-Nowicka, Ewa Harasim-Symbor, Adrian Chabowski

**Affiliations:** Department of Physiology, Medical University of Bialystok, 15-089 Bialystok, Poland; karolina.konstantynowicz@umb.edu.pl (K.K.-N.); eharasim@umb.edu.pl (E.H.-S.); adrian@umb.edu.pl (A.C.)

**Keywords:** insulin signaling, insulin resistance, sphingolipid, lipid accumulation, high-fat diet

## Abstract

Increased lipid bioavailability in a diet favors lipid accumulation, enhancing hepatic lipotoxicity and contributing to insulin resistance (IR) development. The aim of our study was to examine time-dependent alterations in the intrahepatic content of sphingolipids and insulin signaling pathway in rats fed a high-fat diet (HFD). The experiment was conducted on male Wistar rats receiving a standard diet or HFD for five weeks. At the end of each experimental feeding week, liver sphingolipids were determined using high-performance liquid chromatography. The expression of proteins from the sphingolipid pathway and glucose transporter expression were assessed by Western blot. The content of phosphorylated form of proteins from the insulin pathway was detected by a multiplex assay kit. Our results revealed that HFD enhanced hepatic ceramide deposition by increasing the expression of selected proteins from sphingomyelin and salvage pathways in the last two weeks. Importantly, we observed a significant inhibition of Akt phosphorylation in the first week of HFD and stimulation of PTEN and mTOR phosphorylation at the end of HFD. These changes worsened the PI3K/Akt/mTOR signaling pathway. We may postulate that HFD-induced reduction in the insulin action in the time-dependent matter was exerted by excessive accumulation of sphingosine and sphinganine rather than ceramide.

## 1. Introduction

The liver constitutes an organ that plays a pivotal role in lipid and glucose metabolism as well as in the overall regulation of the body’s energy balance [1]. Several studies have demonstrated that nutrient oversupply, especially diets rich in saturated fatty acids, impairs cellular lipid homeostasis and precedes elevated free fatty acid (FFA) level in the bloodstream [1,2]. Increased FFA concentration exceeds hepatic capacity and promotes esterification to the triacylglycerols (TAGs), which are stored then as lipid droplets in the liver [1,3]. Excessive lipid deposition, more than 5% of cell volume, with the absence of profuse alcohol consumption has been defined as non-alcoholic fatty liver disease (NAFLD) [4]. NAFLD is characterized by steatosis with intracellular fat deposition, which may progress to non-alcoholic steatohepatitis (NASH) [1,5,6]. NASH is a more severe form of liver disease where the inflammation process and oxidative stress coexist with excessive lipid accumulation [1,2,5,6]. The next steps of NASH progression are fibrosis and finally cirrhosis and hepatocellular carcinoma (HCC) [5,6].

It is considered that hepatic steatosis development is linked with a disproportion between fatty acid (FA) supply and its expenditure [7]. During simple steatosis development, a dietary pattern is of crucial importance because it may disrupt hepatic lipid homeostasis through the loss of the ability to efficiently metabolize overabundant lipids [8,9,10]. In the condition of high-fat feeding, an excessive lipid provision affects the hepatic lipid metabolism and contributes to the increasing intrahepatic TAG accumulation [2,5]. Furthermore, deposited TAG may be esterified into lipotoxic intermediates from the sphingolipid pathway, especially ceramide (CER) [11,12]. Many studies revealed that excessive ceramide production is tightly related to obesity and simple steatosis deterioration, which predispose to cell ballooning and damage [7,13,14]. Moreover, these molecules not only have pro-apoptotic properties but also impair the insulin signaling pathway [1,7]. A lipid overload condition ceramide exerts an inhibitory effect on activation of protein kinase B (PKB/Akt) mediating in insulin signaling disruption, which favors hyperglycemia and supports insulin resistance (IR) development. What is more, CER impairs glucose entrance into the cell by regulating the membrane translocation of special glucose transporter (GLUT). It is considered that ceramide has a strong influence on glucose metabolism and constitutes an integral factor mediating in liver IR [7]. A growing number of studies have considered that in addition to high-fat diet (HFD), obesity and insulin resistance mediate in the pathogenesis not only NAFLD development but also in its progression into NASH. Moreover, circulating FFAs strongly impair hepatic insulin signaling in response to obesity. In the duration of NAFLD, an increased lipid accumulation in hepatic parenchyma makes glucose metabolism worse and causes suppression of insulin secretion, which promotes IR; hence, the liver degeneration has a connection with changes in peripheral and hepatic insulin resistance development [7].

The interaction between sphingolipid and insulin signaling pathway, during HFD-induced NAFLD development, is to date poorly investigated. In this article, we examined how a time-dependent HFD administration affected the intrahepatic contents of sphingolipids and insulin signaling pathway proteins during NAFLD development and which change appears as first. The discovery which malfunction is the cause and which is the consequence may be clinically relevant in order to find therapeutic target that counteract NAFLD development. On the basis of observed alterations in different time points of high-fat feeding, we may elucidate the “breaking point” in IR development, which predisposes to NAFLD deterioration.

## 2. Results

### 2.1. Effects of Chronic High-Fat Chow Administration on the Hepatic Sphingolipid Concentration

In the liver homogenates, we observed that HFD administration caused a crucial rise in sphinganine (SFA) content only in the third and fifth weeks of examination (+40.9 and +54.2%, respectively; *p* < 0.05; Figure 1A) and a crucial reduction in the fourth week (−25.7%; *p* < 0.05; Figure 1A) compared to the control rats. What is more, at the beginning of HFD, sphinganine content was increased but did not achieve a significant level (*p* = 0.0550; Figure 1A, vs. control group). Chronic high-fat feeding resulted in a pronounced reduction in sphinganine-1-phosphate (SA1P) level only in the third week (−47.1%; *p* < 0.05; Figure 1B). In the case of ceramide, our study showed a notable elevation in ceramide (CER) content in the last two weeks of HFD (fourth week: +28.1%, fifth week: +30.7%; *p* < 0.05; Figure 1C, vs. control group). The high-fat diet substantially elevated the total content of sphingosine (SFO) in all the experimental time points (first week: +73.6%, second week: +41.7%, third week: +40.4%, fourth week: +24.7%, fifth week: +18.2%; *p* < 0.05; Figure 1D). In comparison with the standard chow-fed group, rats from the experimental model exhibited a significantly reduced sphingosine-1-phosphate (S1P) level in the third and fourth weeks (third week: −36.9%, fourth week: −33.3%; *p* < 0.05; Figure 1E). As expected, the value of S1P/CER ratio was diminished by high-fat feeding (second week: −43.0%, third week: −43.8%, fourth week: −48.7%; *p* < 0.05; Figure 1F) in relation to the control group. Additionally, the S1P/CER ratio revealed a trend towards a decrease in the last week of HFD (*p* = 0.0841; Figure 1F).

### 2.2. Effects of Chronic High-Fat Chow Administration on the Hepatic Expression of Proteins from Sphingolipid Pathways

In the liver homogenates, at each week of HFD administration, we observed a markedly decreased expression of proteins involved in ceramide *de novo* synthesis pathway, i.e., serine palmitoyltransferase 2 (SPTLC2; first week: −40.4%, second week: −46.1%, third week: −60.2%, fourth week: −52.8%, fifth week: −58.7%; *p* < 0.05; Figure 2A), ceramide synthase 2 (CerS2; first week: −36.5%, second week: −74.1%, third week: −81.0%, fourth week: −69.1%, fifth week: −70.4%; *p* < 0.05; Figure 2B), and ceramide synthase 6 (CerS6; first week: −47.1%, second week: −41.2%, third week: −68.7%, fourth week: −44.6%, fifth week: −63.1%; *p* < 0.05; Figure 2D). By contrast, the total expression of ceramide synthase 4 (CerS4) increased significantly in the last two weeks of our experiment (fourth week: +82.3%, fifth week: +104.8%; *p* < 0.05; Figure 2C). Concomitantly, we showed a trend towards an increased expression of CerS4 at the beginning of the study (first week: *p* = 0.0842; Figure 2C).

In the case of enzymes that regulate salvage pathway, we observed significant differences in their expression over the course of high-fat chow administration. There is a considerable decrement in sphingosine kinase 2 (SPHK2) expression at every week of the experiment (first week: −51.9%, second week: −72.6%, third week: −78.4%, fourth week: −72.1%, fifth week: −64.0%; *p* < 0.05; Figure 3B). What is more, HFD provoked a rise in the expression of acid ceramidase (ASAH1) only in the second week (+97.7%; *p* < 0.05; Figure 3C) and neutral ceramidase (ASAH2) expression for the whole experiment, except for the third and fifth weeks (first week: +103.9%, third week: +112.6%, fourth week: +94.9%; *p* < 0.05; Figure 3D). Interestingly, at the end of the study, the expression of ASAH2 was significantly decreased (fifth week: −23.4%; *p* < 0.05; Figure 3D) from that in the control group. We also observed a significant decrease in alkaline ceramidase (ASAH3) expression in the first four weeks as a consequence of high-fat feeding (first week: −59.9%, second week: −60.0%, third week: −64.3%, fourth week: −38.7%; *p* < 0.05; Figure 3E). Moreover, the expression of sphingosine kinase 1 (SPHK1) remained unchanged in all the experimental time points (*p* > 0.05; Figure 3A).

In the sphingomyelin pathway, we observed a trend towards to an increase in alkaline sphingomyelinase (Alk-SMase) expression level in the second week of HFD (*p* = 0.0816; Figure 4A). Compared to the control group, the expression of Alk-SMase significantly increased in the third week of experimental feeding (+30.1%; *p* < 0.05; Figure 4A). Additionally, at the end of the experiment only neutral sphingomyelinase (N-SMase) expression was enhanced (fifth week: +27.2%; *p* < 0.05; Figure 4B).

### 2.3. Effects of Chronic High-Fat Chow Administration on the Hepatic Expression or Phosphorylation State of Proteins Involved in the Insulin Signaling Pathway

In the liver homogenates, a high-fat feeding provoked a significant diminished in the level of phosphorylated phosphatase and tensin homolog (pPTEN(Ser380)) at the beginning of our study (first week: +11.3%, second week: +12.7%; *p* < 0.05; Figure 5B). Additionally, in the last week of the experimental feeding, we noticed a trend towards to an increase in pPTEN (Ser380) level (fifth week: *p* = 0.0704; Figure 5B). We also observed a decrease in phosphorylated protein kinase B (pAkt(Ser473)) content in almost all the weeks of HFD (first week: −27.3%, second week: −22.4%, third week: −38.4%, fourth week: −15.5%; *p* < 0.05; Figure 5C) and in phosphorylated glycogen synthase 3 α/β (GSK3α/β(Ser21/Ser9)) content in all the experimental weeks (first week: −41.4%, second week: −43.0%, third week: −40.2%, fourth week: −46.3%, fifth week: −20.2%; *p* < 0.05; Figure 5D) in comparison with rats fed a standard chow. Moreover, level of phosphorylated mechanistic target of rapamycin (pmTOR(Ser2448)) protein was reduced at the beginning of the study (first week: −21.5%, second week: −18.2%, third week: −25.7%, fourth week: −16.8%; *p* < 0.05; Figure 5F), and was then enhanced in the last weeks of high-fat chow administration (fifth week: +8.5%; *p* < 0.05; Figure 5F). In our study, we also noted a significant increase in phosphorylated ribosomal protein S6 kinase (pP70 S6 kinase(Thr389)) level and decrease in phosphorylated S6 ribosomal protein (pS6RP(Ser235/Ser236)) level throughout the whole experiment (pP70 S6 kinase (Thr389)—first week: +21.5%, second week: −22.6%, third week: +28.8%, fourth week: +17.8%, fifth week: −29.8%; *p* < 0.05; Figure 5G and pS6RP (Ser235/Ser236)—first week: −78.6%, second week: −70.8%, third week: −80.7%, fourth week: −64.1%, fifth week: −35.3%; *p* < 0.05; Figure 5H). On the other hand, we did not observe a significant change in the fluorescent intensity in the phosphorylated insulin receptor substrate (pIRS(Ser636/Ser639)) and phosphorylated Bcl-2-associated agonist of cell death (pBAD(Ser136)) (*p* > 0.05; Figure 5A,E, respectively; vs. control group).

Moreover, we noticed a diminishment in the expression of glucose transporter 2 (GLUT2) in every week of the experimental feeding (first week: −45.2%, second week: −50.3%, third week: −68.3%, fourth week: −58.2%, fifth week: −61.0%; *p* < 0.05; Appendix A) in comparison with rats from the control group.

## 3. Discussion

The current study aims to elucidate the role of ceramide and its derivatives from the sphingolipid pathway in the development of insulin resistance during the time course of HFD in rats.

Studies conducted on animal models showed that an enhanced supply of FAs in the diet provoked an increase in ceramide deposition in the liver as well as other than liver peripheral tissues, e.g., cardiac and skeletal muscles [15,16,17]. In a study provided by Zabielski et al., already after 7 days from the beginning of the HFD treatment, rats’ liver tissue was characterized by increased CER deposition [18]. Similar observations were presented by Chocian et al., where the 3 weeks of HFD induced a threefold increase in hepatic accumulation of ceramide [12]. However, in our study, we noticed a significant enhancement in CER content in the last two weeks of HFD (fourth and fifth weeks of the experiment). This slight discrepancy may arise from different hepatocytes’ abilities to “buffer” FA and enhance their storage as cells’ adaptation to the condition of excessive FA availability [9,10].

Ceramide is a key mediator of lipotoxicity resulting from the excessive delivery of saturated fatty acids in the diet as substrates for the rate-limiting *de novo* synthesis pathway of ceramide [19]. On the contrary, in our study, we demonstrated that enzyme from the abovementioned pathway–SPTLC2 and two enzymes that share *de novo* and salvage pathways, i.e., CerS2 and CerS6 were depressed. However, a nearly 40% increase in SFA content was noted in the third and fifth weeks of experimental feeding. These observations may be partially explained by a simultaneous decrease in the SA1P level in the third week of HFD, which might be a possible way for converting SA1P to SFA in the liver. It is possible that in such a short time of HFD feeding as we used in this study, the *de novo* pathway is not activated, and hence we observed a weakening of this pathway. On the other hand, we observed an intensification of ceramide formation mostly through the regeneration pathways (the salvage and sphingomyelin pathways), which is responsible for the synthesis of sphingolipids from 50% to 90% [20]. The significant elevation in the ceramide content may be the outcome of the increased breakdown of sphingomyelin, for which increased concentration was observed at the first two weeks of HFD feeding in our previous study [3]. This assumption was supported by the increased hepatic expression of Alk-SMase and N-SMase in the third and fifth weeks, respectively, in the results presented herein. It should be noted that ceramide formed from sphingomyelin as well as synthetized in the *de novo* pathway may be then hydrolyzed to sphingosine [20]. Importantly, lipidomic profiling showed that not only ceramide but also other sphingolipids, e.g., S1P or SFO, correlated with hepatic insulin resistance development [19,21]. In this study, we also observed an elevation in the content of SFO at every time point of experimental feeding. It is particularly important that ceramide regeneration by a breakdown of sphingosine bases may be even more destructive and more strongly inhibit insulin signaling pathway than CER formed by *de novo* pathway [22]. Increased SFO level was supported by increased expression of selected ceramidases, especially neutral. Interestingly, changes in CER and SFO concentrations with a diminishment in the expression of CerS2 and CerS6, as well as increment in the expression of hepatic-specific ASAH2 in the whole feeding period may suggest that the formation of SFO from CER was more intensified compared to the regeneration of CER from SFO. Thus, we suspect that in our research during high-fat feeding, the salvage pathway is mainly responsible for the increased ceramide accumulation. Moreover, we found the decreased S1P content in the liver of HFD-fed rats and simultaneously decreased the liver-specific SPHK2 expression. Mentioned effects in S1P level and SPHK2 expression were the most severe in the third and fourth weeks of high-fat feeding, suggesting that in these weeks, there is a significant inhibition in the catabolism pathway, which predisposed to the development of simple hepatic steatosis [7]. A similar effect was observed in mice fed an HFD for 4 weeks, which revealed a significant downregulation in either mRNA and protein levels of SPHK2 that caused the activation of oxidative genes deteriorating hepatosteatosis and affecting associated hepatic glucose metabolism [23]. In addition, SPHK2 knockout mice characterized rapid development of steatosis after 2 weeks of HFD, pointing out that SPHK2 and also generated S1P can determine the activity of enzymes and receptors that regulate the lipid metabolism [24]. Sphingosine-1-phosphate and ceramide have an opposite influence on cells’ metabolism and survival. S1P increases glucose uptake and metabolism as well as promotes cell survival and proliferation, while CER inhibits glucose metabolism, amplifies the inflammatory response, and promotes cell death [7,25]. The interaction between the content of S1P and CER should be supported by alternations in S1P/CER balance that clarify which sphingolipid, S1P or CER, plays a pivotal role in regulating cell biology related to hepatic injury induced by HFD. This assumption was confirmed by decreased the value of S1P/CER ratio in nearly every week of experimental feeding in our present study. Thus, we may suggest that a decrease in S1P and an increase in CER along with diminishment in S1P/CER ratio were partially mediated by the accumulation of bioactive lipids and associated with the occurrence of hepatic IR.

The mechanism of sphingolipid species inducing hepatic insulin resistance has not been completely explained. We presented that HFD directly attenuated insulin signaling transduction pathway. In skeletal muscle, increased ceramide deposition induced insulin resistance development [26]. Surprisingly, in our study, enhanced accumulation not of ceramide but of other sphingolipid species significantly reduced the level of phosphorylated Akt at Ser473 already in the first week of administration of chow rich in fat and was continued in the next weeks of our experimental feeding. Our observation in NAFLD rats is consistent with numerous studies on a rat model where different times of high-fat feeding (10, 12, and 24 weeks of HFD) caused a reduction in the hepatic level of phosphorylated Akt (Ser473) [27,28,29]. The most likely explanation of the abovementioned correlation is decreased expression of CerS2 (the lowest in the third week of our study) that promotes C16-ceramide generation, which strongly inhibits Akt phosphorylation (the lowest pAkt level was observed also in the third week of HFD) and aggravates insulin signaling in the liver [30,31]. In response to the diminishment in the phosphorylation of Akt (Ser473), the total expression of GLUT2 was abolished already in the first week of HFD. In this context, we supposed that decreased expression of glucose transporter in the hepatocytes altered its ability for glucose sensing due to decreased GLUT2 membrane localization, which reduced hepatic glucose uptake and may be strongly associated with development of hyperglycemia (data presented in our previous study [32]). Moreover, enhanced blood glucose concentrations may favor increased accumulation of saturated fatty acid supplied with the chow rich in fat that finally promoted development of liver steatosis and deterioration to NASH [33]. In addition, we also found that at the beginning of our study, the content of phosphorylated PTEN at residue Ser380, as a pivotal regulator of PIP3 (phosphatidylinositol-3,4,5-triphosphate) degradation in the PIP3 kinase (PI3K) pathway, was enhanced upon experimental feeding. The altered phosphorylation resulted in decreased activation of PTEN, leading to inhibited Akt-signaling pathway and decreased hepatic insulin sensitivity [34]. Studies conducted on the liver samples and primary hepatocytes from rats treated with a high-caloric diet for 12 and 15 weeks, respectively, also showed the increased pPTEN levels [35,36]. Moreover, Kohil et al. reported that the short-term (18 h) hepatocytes’ exposition to a high concentration of long-chain fatty acids caused inhibition of the PI3K pathway *via* maintaining a high level of pPTEN [37]. These findings are in line with our observations where rats fed a HFD showed reduced pAkt and simultaneously enlarged pPTEN levels that are responsible for impaired insulin action in the liver tissue [35]. We showed that enhanced level of SFO (not CER) already in the first week of HFD administration, continued in each experimental week is reflected by changes in insulin pathway proteins. The abovementioned development of disruption in the insulin action induced by SFO was probably enhanced by increased SFA deposition mainly in the third week of HFD and also by increased CER accumulation at the end of study. The most important finding from our study is that in the liver excessive accumulation of other sphingolipid fractions such as SFO may induce insulin resistance development in early stages of steatosis. The main weakness of our study is that we did not use a selective inhibitor of CerSs in the *de novo* and salvage pathways such as fumonisin B1 for clear confirmation as to which pathway is mainly responsible for observed changes in sphingolipid content [20,38].

The diminishment in the level of pAkt depressed the phosphorylation of mTOR at residue Ser2448 at the beginning of our experimental feeding and was continued for four weeks of HFD. However, in the last week, we observed a significant increase in the pmTOR level in the liver from NAFLD rats. According to the research by Wang et al., a significant enhancement in the hepatic pmTOR protein expression was observed in rats after 8 and 16 weeks of HFD exposition, corresponding to the progression of simple steatosis into hepatitis. Interestingly, the expression after 16 weeks of HFD was higher than after 8 weeks of HFD, which might reveal an enhanced activity of the mTOR complex that regulates nutrition transport and may disclose insulin sensitivity deterioration and NAFLD progression into NASH [39]. A discrepancy between the results of Wang et al., those of Khamzina et al., and our own results on changes in the pmTOR level may be explained by the different duration of HFD. We paid attention to only 5 week chow administration, and we suppose that a longer time of HFD (8 or 16 weeks) may cause a significant increase in the phosphorylated mTOR level. Following a high-fat diet administration, we also noticed elevated Thr389 P70S6 kinase phosphorylation in the liver. Supporting these data is research presented by Khamzina et al., where mTOR activated P70S6 kinase isoform 1 by its phosphorylation (approximately 20-fold increase) in obese rats with IR. Moreover, an increase in the phosphorylation of mTOR (Ser2448) and P70S6 kinase (Thr389) may be a consequence of hyperinsulinemia, which exacerbates the phosphorylation state [40]. In line with the abovementioned findings, the experiment carried out on HepG2 cells revealed an increased level of pmTOR and pP70S6 kinase after short-time insulin exposition [40], which suggested the pivotal role of the mTOR complex in disrupting hepatic insulin signaling. It is worth pointing out that our dietary model already in the first week of HFD caused a dramatic (fourfold) decrease in the hepatic phosphorylation of ribosomal protein S6 (Ser235/Ser236), the protein whose phosphorylation leads to upregulated translation of proteins and is also linked with hepatocytes proliferation [41,42]. It is well known that the phosphorylation of S6RP is regulated by two isoforms of S6 kinases—P70 and P90. The observed decrease in pS6RP (Ser235/Ser236) may be a result of the decreased pP90S6 kinase level rather than increased P70S6 kinase phosphorylation, leading to the abolishment in the phosphorylation of the ribosomal protein [43]. We also suspect that the diminishment in the level of pS6RP may reflect an inhibitory effect on the cells proliferation process that subsequently may cause development of apoptotic changes in the hepatocytes at the beginning of HFD administration. However, in our study, we did not observe a significant increase in BAD phosphorylation. The most possible explanation may be the fact that protective mechanisms in the liver were activated. Thus, future studies should be focused on searching new methods or substances that would stimulate protective mechanisms of this tissue.

## 4. Materials and Methods

### 4.1. Experimental Model

The animal experiment was performed in accordance with the institutional ethical guidelines, Ethics Committee on Animal Care at the Medical University of Bialystok (28 May 2008; approval number: 32/2008). Male Wistar rats with initial body weight of 100–150 g were subjected to the standard facilitating conditions (on a reverse 12 h light/dark cycle, at 22 ± 2 °C and ad libitum access to water and standard rodent chow) for 7 days to adapt to the laboratory surroundings. Subsequently, rats were randomly selected into two groups according to the type of diet, namely, (1) a control group (six rats in each group) fed with a standard rodent chow with content of a reduced fat-calorie (kcal of energy distribution: 57.1% carbohydrates, 30.5% protein, and 12.4% fat; Σ saturated: 36.0%, Σ unsaturated: 64.0%) or (2) a high-fat diet group (six rats per group in each week of the diet) fed with a chow rich in fatty acids purchased from New Brunswick (NJ, USA, cat no. D12492; kcal of energy distribution: 60% fat, 20% protein, and 20% carbohydrates; Σ saturated: 72.0%, Σ unsaturated: 28.0%). The fatty acid composition of the experimental high-fat diet was measured by gas–liquid chromatography (GLC). At the beginning of the experiment, we had two control groups: the first in the first week and the second in the fifth week, both receiving a standard diet. In the absence of significant changes between the two control groups, the results were referred to the first week control group (called 0 week). All the animals, after overnight fasting, at the end of the different feeding time points (first, second, third, fourth, and fifth weeks) were anaesthetized by the intraperitoneal pentobarbital injection (80 mg/kg of body mass). Then, rats were euthanized by bleeding, and the liver was promptly prepared and frozen by the use of pre-cooled aluminum tongs in liquid nitrogen. The tissues were harvested and stored at −80 °C until measurements.

### 4.2. Immunoblotting

Before the Western blot procedure, all the liver tissues were homogenized in radioimmunoprecipitation assay buffer (RIPA) containing protease and phosphatase inhibitor cocktail (Roche Diagnostics GmbH, Mannheim, Germany). The homogenates were centrifuged at 10,000× *g* at 4 °C for 30 min, and after that, in obtained supernatants using bicinchoninic acid (BCA) with bovine serum albumin (BSA) as a standard, the total protein concentrations were determined. All the samples of the liver tissue (30 μg of proteins) were reconstituted in Laemmli Sample Buffer and separated by 10% sodium dodecyl sulfate–polyacrylamide gel electrophoresis (SDS-PAGE; Bio-Rad, Hercules, CA, USA). After electrophoresis, samples were transferred onto polyvinylidene fluoride (PVDF) or nitrocellulose membranes. The membranes were blocked in tris-buffered saline with Tween-20 (TBST) and 5% BSA or non-fat dry milk and then immunoblotted overnight with primary antibodies: serine palmitoyltransferase 2 (SPTLC2; 1:500, sc-27500, Santa Cruz Biotechnology, Inc., Dallas, TX, USA), ceramide synthase 2 (CerS2; 1:500, ab85567, Abcam, Cambridge, United Kingdom), ceramide synthase 4 (CerS4; 1:500, ab66512, Abcam, Cambridge, United Kingdom), ceramide synthase 6 (CerS6; 1:500, ab77603, Abcam, Cambridge, United Kingdom), sphingosine kinase 1 (SPHK1; 1:500, SAB2500984, Sigma Aldrich, Saint Louis, MO, USA), sphingosine kinase 2 (SPHK2; 1:500, SAB4502433, Sigma Aldrich, Saint Louis, MO, USA), acid ceramidase (ASAH1; 1:500, sc-28486, Santa Cruz Biotechnology, Inc., Dallas, TX, USA), neutral ceramidase (ASAH2; 1:200, sc-68906, Santa Cruz Biotechnology, Inc., Dallas, TX, USA), alkaline ceramidase (ASAH3; 1:500, PA5-75603, Thermo Fisher Scientific, Inc., Waltham, MA, USA), alkaline sphingomyelinase (Alk-SMase; 1:500, sc-49352, Santa Cruz Biotechnology, Inc., Dallas, TX, USA), neutral sphingomyelinase (N-SMase; 1:500, sc-26214, Santa Cruz Biotechnology, Inc., Dallas, TX, USA), and glucose transporter 2 (GLUT2; 1:200, sc-7580, Santa Cruz Biotechnology, Inc., Dallas, TX, USA). The membranes were incubated with suitable secondary horseradish peroxidase (HRP)-conjugated antibodies (1:3000, 7074S, Cell Signaling Technology, Danvers, MA, USA; 1:3000, sc-516102, Santa Cruz Biotechnology, Inc., Dallas, TX, USA; 1:3000, sc-2354, Santa Cruz Biotechnology, Inc., Dallas, TX, USA). Next, the protein bands in the liver homogenates were visualized using the chemiluminescence substrate (Clarity Western ECL Substrate; Bio-Rad, Hercules, CA, USA) and then immunoblotting signals were quantified densitometrically and visualized by a ChemiDoc system (Bio-Rad, Hercules, CA, USA). The obtained proteins signals were standardized to the total expression of protein. The results are expressed in percentages of the control.

### 4.3. Sphingolipids Analysis

The sphinganine (SFA), sphinganine-1-phosphate (SA1P), ceramide (CER), sphingosine (SFO), and sphingosine-1-phosphate (S1P) contents were analyzed using the high-performance liquid chromatography (HPLC) method according to the previously described protocol by Hodun et al. [44]. Briefly, the liver tissue was homogenized. Then, in the presence of internal standards (C17-sphingosine-1-phosphate and C17-sphingosine, Avanti Polar Lipids), lipids from homogenates were ultrasonicated and extracted by the addition of chloroform. The sphingoid base-1-phosphates were measured after dephosphorylation to sphinganine and sphingosine with the addition of alkaline phosphatase (Sigma Aldrich, Saint Louis, MO, USA). Next, a small aliquot was transferred into a fresh tube containing N-palmitoyl-D-erythro-sphingosine (C17-base) as an internal standard. The samples were next subjected to convert CER into SFO by alkaline hydrolysis. Dephosphorylated sphingoid bases, free sphingosine, and sphinganine, as well as sphingosine sourced from ceramide were converted to their o-phthalaldehyde derivatives and analyzed with the use of the HPLC system (Varian ProStar, Agilent Technologies, Santa Clara, CA, USA) equipped with a fluorescence detector and C18 reversed-phase column (OmniSpher 5, Varian Inc. 4.6 × 150 mm).

### 4.4. Intracellular Content of Phosphoproteins

Before the Bio-Plex Pro Cell Signaling Assay procedure, the lysates from the liver samples were prepared in adequate cell lysis buffer (Bio-Rad, Hercules, CA, USA) with the addition of phenylmethylsulfonyl fluoride (PMSF; Sigma Aldrich, Saint Louis, MO, USA) and cell lysis factor QG (Bio-Rad, Hercules, CA, USA). Then, the obtained lysates were centrifuged at 15,000× *g* at 4 °C for 10 min. In these samples’ supernatant, the total protein concentration was measured. The ultimate protein concentration of samples was in the range of 20–200 µg/mL. Finally, the supernatants were recollected and stored at −80 °C for further analysis.

According to the protocol, the content of phosphorylated form of analytes was detected by multiplex assay kits employing covalently coupled magnetic beads. Briefly, a 96-well plate was prewashed two times with a wash buffer. Then, all the samples and blank and cell lysis controls were added to the appropriate wells and incubated in the dark overnight on a plate shaker. The second day, after three times washing with a wash buffer, detection antibodies were added to the wells for 30 min of incubation in the dark. Following the addition of streptavidin–phycoerythrin (SA-PE) conjugate, another incubation, and a series of washes, the resuspend beads were added and shook for 30 s. Finally, the plate was placed on the reader and a relative concentration of analytes was determined by the use of the Bio-Plex 200 system (Bio-Rad).

### 4.5. Data Analysis

The results are presented as the mean ± standard deviation (SD) on the basis of the six independent determinations in each examined group. The data were analyzed using GraphPad Prism software version 5 (La Jolla, CA, USA). Then, the Shapiro–Wilk test and Bartlett’s test were applied to ensure normal distribution and uniformity of the results. Statistical differences between the study groups were analyzed by one-way test ANOVA followed by an appropriate post hoc test (Tukey’s test and *t*-test). Statistically significant differences were presented as a *p*-value < 0.05.

## 5. Conclusions

The present study provided a novel insight into the development of insulin resistance at the different time points of high-fat feeding during NAFLD occurrence. Most importantly, an excessive accumulation of sphingosine caused disruption in the insulin action already in the first week of experimental feeding. This effect on the insulin transduction pathway was exacerbated in the last three weeks of HFD by increased SFA and CER levels. Our findings clearly demonstrated that only at the end of feeding, increased ceramide accumulation sustained insulin resistance rather than it being an inducer of its development. We suspect that the main route through which CER was deposited was enhancing sphingomyelin and salvage pathways activity with simultaneous inhibition of CER catabolism. This study demonstrated that the PI3K/Akt/mTOR pathway is strongly activated depending on the high-fat feeding time, particularly in the two last weeks of 5 week HFD (Figure 1). We believe that these results indicated the precise time point when HFD caused changes in the sphingolipid metabolism that affected the insulin sensitivity disturbances. The presented findings are very important because they may allow to find a potential therapeutic target in the early stage of liver disease development.

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
