# Peer review of "Time-Dependent Changes in Hepatic Sphingolipid Accumulation and PI3K/Akt/mTOR Signaling Pathway in a Rat Model of NAFLD"

_ijms, 2021, doi:10.3390/ijms222212478_

Round 1

Reviewer 1 Report

In the manuscript “Time-Dependent Changes in the Hepatic Sphingolipids Accumulation and PI3K/Akt/mTOR Signaling Pathway in NAFLD Rats”, the authors proposed to investigate how a time-dependent HFD administration affected the intrahepatic contents of sphingolipid and insulin signaling pathway proteins during NAFLD development. The paper could be of interest but there are too many drawbacks that hampered the initial enthusiasm. All sections need revision and the paper should be improved. I have several suggestions.

Specific comments:

  1. Title: needs some adjustments: “… in NAFLD rats” is somewhat confusing please make some changes as “…in rats with NAFLD” or just use the term “rat model of NAFLD”.
  2. Abstracts needs extensive revision: what strain of rats was used? How many rats? What is the diet composition? Please include relevant information for the paper. This is a paper mostly based on NAFLD model thus, information regarding it must be highlighted.
  3. The rationale for the experiments is not clear. The should better explain their experimental approach and the relevance.
  4. Discussion needs major revision. It is unclear the “take home message” or the biological/clinical relevance of the study. Please highlight the major results and answer to your experimental question with a clear contribution to the literature.
  5. Several methodological aspects are missing. For instance, antibodies catalog number and dilutions should be included.

Author Response

In the manuscript “Time-Dependent Changes in the Hepatic Sphingolipids Accumulation and PI3K/Akt/mTOR Signaling Pathway in NAFLD Rats”, the authors proposed to investigate how a time-dependent HFD administration affected the intrahepatic contents of sphingolipid and insulin signaling pathway proteins during NAFLD development. The paper could be of interest but there are too many drawbacks that hampered the initial enthusiasm. All sections need revision and the paper should be improved. I have several suggestions.

Specific comments:

  1. Title: needs some adjustments: “… in NAFLD rats” is somewhat confusing please make some changes as “…in rats with NAFLD” or just use the term “rat model of NAFLD”.

Authors: We adjusted the title of our manuscript to the following: “Time-Dependent Changes in the Hepatic Sphingolipids Accumulation and PI3K/Akt/mTOR Signaling Pathway in Rat Model of NAFLD”.

  1. Abstracts needs extensive revision: what strain of rats was used? How many rats? What is the diet composition? Please include relevant information for the paper. This is a paper mostly based on NAFLD model thus, information regarding it must be highlighted.

Authors: We understand that Abstract does not include all relevant information about the experimental procedure and therefore may seem to be not precise. However, the total amount of Abstract words should be about 200 as recommended by the Journal. We described in detail the experimental conditions in the main text of our manuscript in the Experimental model in the Materials and Methods section.

  1. The rationale for the experiments is not clear. The should better explain their experimental approach and the relevance.

Authors: Thank you for this comment. We added information to the Introduction section in order to clarify the main goal of or study and also the aim into the Abstract. Lines: 66-74.

  1. Discussion needs major revision. It is unclear the “take home message” or the biological/clinical relevance of the study. Please highlight the major results and answer to your experimental question with a clear contribution to the literature.

Authors: We moderated some aspects in the Discussion highlighting the main results of experimental study and relation between changes in the sphingolipid species and insulin signaling pathway, which was observed in our study. Moreover, we answered the major question about aim of this study which was concluded in the Conclusions part. Lines: 265-272, 299-309, 450-466.

  1. Several methodological aspects are missing. For instance, antibodies catalog number and dilutions should be included.

Authors: In the Materials and Methods section, we added more information about the Western Blot procedure, e.g., catalog number and dilutions of primary and secondary antibodies. Lines: 383-399.

Reviewer 2 Report

The aim of this study was to evaluate the time-dependent effects of high fat diet on the intrahepatic contents of sphingolipid and insulin signaling in the liver in male rats. Overall, manuscript is well-written and interesting, however I have several comments to the authors:

  1. Abstract lacks the aim of the study. This needs to be added.
  2. It seems to be that first sentence is incomplete. Do the authors mean glucose and lipid metabolism or homeostasis?
  3. It is disappointing the manuscript lacks data demonstrating the effects of HFD on body weight and metabolic parameters such glucose and lipid profile. These results could be included in manuscript at least as supplementary data.
  4. Western blot. Cat. no. and dilution of all antibodies should be provided.
  5. Protein production. Can authors explain why they did not use loading control such as GAPDH or beta actin? What is the total protein expression shown in figures? How was it stained, detected and evaluated? These points should be clarified.
  6. Data presenting the effects of HFD on GLUT2 protein production are not mentioned and elaborated in the discussion.
  7. The authors can improve discussion by pointing out limitations of this work.
  8. Scheme 3 should be briefly descripted.
  9. In my opinion the authors should provide a list of abbreviations used in manuscript.

Author Response

The aim of this study was to evaluate the time-dependent effects of high fat diet on the intrahepatic contents of sphingolipid and insulin signaling in the liver in male rats. Overall, manuscript is well-written and interesting, however I have several comments to the authors:

  1. Abstract lacks the aim of the study. This needs to be added.

Authors: We agree with the Reviewer and we added the aim of our study to the Abstract.

  1. It seems to be that first sentence is incomplete. Do the authors mean glucose and lipid metabolism or homeostasis?

Authors: Indeed, the first sentence is incomplete. We changes it in sentence to “The liver constitutes an organ, which plays a pivotal role in lipid and glucose metabolism as well as in the overall regulation of the body’s energy balance.” Lines: 29-30.

  1. It is disappointing the manuscript lacks data demonstrating the effects of HFD on body weight and metabolic parameters such glucose and lipid profile. These results could be included in manuscript at least as supplementary data.

Authors: We understand that basic parameters during the experiment are important, but in our previous studies conducted on the same animal model we have already demonstrated the effects of high-fat feeding on the basic metabolic parameters such as plasma glucose and insulin levels, HOMA-IR, and also lipids content in the liver and plasma:

Int J Mol Sci. 2019 Aug 5;20(15):3829. doi: 10.3390/ijms20153829. High-Fat Feeding in Time-Dependent Manner Affects Metabolic Routes Leading to Nervonic Acid Synthesis in NAFLD. Karolina Konstantynowicz-Nowicka, Klaudia Berk, Adrian Chabowski, Irena Kasacka, Patrycja Bielawiec, Bartłomiej Łukaszuk, Ewa Harasim-Symbor.

Cell Physiol Biochem. 2015;37(3):1147-58. doi: 10.1159/000430238. Epub 2015 Sep 25. Myocardial lipid profiling during time course of high fat diet and its relationship to the expression of fatty acid transporters. Ewa Harasim, Tomasz Stępek, Karolina Konstantynowicz-Nowicka, Marcin Baranowski, Jan Górski, Adrian Chabowski.

  1. Western blot. Cat. no. and dilution of all antibodies should be provided.

Authors: In the Materials and Methods section, we added more information about the Western Blot procedure, e.g., catalog number and dilutions of primary and secondary antibodies. Lines: 383-399.

  1. Protein production. Can authors explain why they did not use loading control such as GAPDH or beta actin? What is the total protein expression shown in figures? How was it stained, detected and evaluated? These points should be clarified.

Authors: There are two methods of Western blot normalization. One is the use of housekeeping protein as was suggested by the Reviewer. However, recent studies indicated that expression of housekeeping proteins is not stable and may be altered by some experimental conditions. Thus, in our experiment we selected other normalization method namely normalization by sum using total protein staining. The lower panel on each graphs (Figure 2, 3, 4 as well as Figure 1 in Supplementary Material) is the image of Stain-Free labeled proteins after membrane transfer that is used for normalization. The Stain-Free system from Bio-Rad that was used in our study takes chemical labeling of the sample proteins. The labeling moiety, a trihalo compound, is incorporated into the proprietary pre-cast gel. When this gel is exposed to UV light, the trihalo label covalently binds tryptophan residues in the sample proteins and forms a cross-linked fluorescent product. Fluorescently-modified proteins are then detected with a CCD imager and UV illumination. Next, the image of total protein and protein of interest overlap in ImageLab system that is attached to ChemiDoc visualization system. Total protein staining after membrane transfer is possibly the most reliable and accurate method for normalization of Western blot data:

Analytical Biochemistry, Volume 440, Issue 2, 15 September 2013, Pages 186-188: Stain-Free total protein staining is a superior loading control to β-actin for Western blots. Jennifer E.Gilda, Aldrin V.Gomes.

We added more information to the Materials and Methods such as the amount of total protein (30 μg for each sample), which loaded in each lane on the electrophoresis gels and the type of visualization of protein bands. Moreover, below the Figures demonstrated proteins expression, we have attached information about the manner presentation of data (the percentage difference presented vs. control group which was set as 100%).

  1. Data presenting the effects of HFD on GLUT2 protein production are not mentioned and elaborated in the discussion.

Authors: We added information about alternations in the expression of GLUT2 during experimental feeding. In the part of Discussion, we also explained the correlation between its expression and the development in insulin resistance and steatosis induced by high-fat diet administration. Lines: 279-287.

Figure presented GLUT2 expression was added to Supplementary Materials.

  1. The authors can improve discussion by pointing out limitations of this work.

Authors: We added the limitations of our study. Lines: 306-309.

  1. Scheme 3 should be briefly descripted.

Authors: We apologize for the mistake, this Scheme should be name “Scheme I”. We have corrected in the scheme signature. This Scheme I constitutes a graphical summary of our results, so we decided to cite the scheme in the Conclusions section. As suggested we slightly expanded the signature under the diagram. Lines: 462, 468-469.

  1. In my opinion the authors should provide a list of abbreviations used in manuscript.

Authors: Abbreviations added. Lines: 488-533.

Round 2

Reviewer 1 Report

The revised version has improved. 

Reviewer 2 Report

Thank you for addressing my comments, manuscript was adequately improved.